# Endoscopic Ultrasound-Guided Fine Needle Biopsy in the Diagnostic Work-Up of Deep-Seated Lymphadenopathies and Spleen Lesions: A Monocentric Experience

**DOI:** 10.3390/diagnostics13172839

**Published:** 2023-09-01

**Authors:** Flaminia Bellisario, Fabia Attili, Fabrizia Campana, Federica Borrelli de Andreis, Silvia Bellesi, Elena Maiolo, Eleonora Alma, Rosalia Malafronte, Giuseppe Macis, Luigi Maria Larocca, Salvatore Annunziata, Francesco D’Alò, Stefan Hohaus

**Affiliations:** 1Dipartimento di Diagnostica per Immagini, Radioterapia Oncologica ed Ematologia, Fondazione Policlinico Universitario A. Gemelli IRCCS, 00168 Rome, Italy; flaminia.bellisario@gmail.com (F.B.); silvia.bellesi@policlinicogemelli.it (S.B.); elena.maiolo@policlinicogemelli.it (E.M.); eleonora.alma@gmail.com (E.A.); rosalia.malafronte@unicatt.it (R.M.); giuseppe.macis@unicatt.it (G.M.); salvatore.annunziata@policlinicogemelli.it (S.A.); stefan.hohaus@unicatt.it (S.H.); 2Sezione di Ematologia, Dipartimento di Scienze Radiologiche ed Ematologiche, Università Cattolica del Sacro Cuore, 00168 Rome, Italy; fabrizia.campana@gmail.com; 3Endoscopia Digestiva, Dipartimento di Scienze Mediche e Chirurgiche, Fondazione Policlinico Universitario A. Gemelli IRCCS, 00168 Rome, Italy; fabia.attili@policlinicogemelli.it (F.A.); federica.bda@gmail.com (F.B.d.A.); 4Patologia Oncoematologica, Dipartimento di Scienze della Salute della Donna, del Bambino e di Sanità Pubblica, Fondazione Policlinico Universitario A. Gemelli IRCCS, 00168 Rome, Italy; luigimaria.larocca@policlinicogemelli.it

**Keywords:** endoscopic ultrasound, fine needle biopsy, flow cytometry, lymphoma

## Abstract

EUS-FNB has been introduced in clinical practice as a less invasive diagnostic approach with respect to surgery. We performed a single-center retrospective study on the diagnostic efficacy of EUS-guided FNB, including 171 patients with lymph nodes, splenic, and extranodal lesions that underwent EUS for FNB at our institution. Excluding 12 patients who did not undergo FNB and 25 patients with a previous diagnosis of a solid tumor, we included 134 patients with clinical/radiological suspect of a lymphoproliferative disease, including 20 patients with a previous history of lymphoma. Out of the 134 biopsies, material of diagnostic quality was obtained in 111 procedures (84.3%). Histological examination of the EUS-FNB samples produced an actionable diagnosis in 100 cases (74.6%). Among the patients without an actionable diagnosis, a second, different diagnostic procedure produced a further eight diagnoses of lymphoma. Therefore, the sensitivity of EUS-FNB for diagnosing lymphomas was calculated to be 86.4% (51/59). Assignment of lymphomas to WHO classification subtypes was possible in 47/51 (92%) of the cases. In conclusion, EUS-FNB is an effective procedure for the histological characterization of lesions that are suspected to be lymphoproliferative disease, allowing for an actionable diagnosis in 75% of cases.

## 1. Introduction

Diagnosing lymphoma relies on evaluating samples for morphological examination, immunophenotypic analysis, genetic profiling, and molecular characterization. However, obtaining these samples can be challenging, depending on their location [1]. In cases where the lymph node or pathological extra-nodal tissue is superficially located, an excisional biopsy is recommended. However, for deep-seated lesions, a surgical biopsy may be overly invasive and requires general anesthesia. Alternative approaches, such as percutaneous computed tomography (CT) or ultrasound (US)-guided fine-needle biopsies (FNB), as well as endoscopy–ultrasound (EUS)-guided FNB, have shown good diagnostic efficiency [2,3].

EUS-FNB has been utilized in clinical practice for over 20 years and has become a routine procedure in many hospitals [4,5]. Combining digestive endoscopy with ultrasound imaging, this technique enables a close proximity to lesions surrounding the gastrointestinal tract, including the bilio-pancreatic area, masses, and lymph nodes. The instrument allows for obtaining multiple samples for histology, cytology, flow cytometry, and cultural assays with minimal invasiveness, as it passes a needle through the layers of the gastrointestinal tract.

In recent years, the application of EUS-FNB has been validated in various diagnostic and therapeutic algorithms, primarily for pancreatic or gastrointestinal malignancies [6,7]. It is a procedure that is considered safe, although with possible complications that are estimated to occur in 1 to 2% of patients. Pain, acute pancreatitis, infection, and bleeding are the primary adverse events [8].

EUS also allows for sampling from mediastinal, retroperitoneal, and perigastrointestinal lymph nodes. The overall accuracy has been reported to be between 65% and 100% [9,10]. However, the role of EUS-FNB in diagnosing lymphadenopathy remains uncertain, and there is a dearth of high-quality, prospective studies on this subject [11,12]. 

To address the current application of EUS in the diagnostic work-up of deep lymphadenopathies and spleen lesions in cases of suspected lymphoproliferative diseases, we conducted a retrospective, monocentric study to evaluate the efficacy of biopsies performed using this technique.

## 2. Materials and Methods 

The study retrospectively analyzed EUS-guided biopsies of deep lymph nodes, spleen, and extranodal lesions performed at our institution from June 2017 to December 2021 for suspected lymphoma or malignant lymph nodes. 

The EUS-FNB technique combines EUS imaging with fine-needle tissue acquisition to obtain tissue samples from targeted areas. The procedure was performed using a disposable standard needle. An Acquire TM Endoscopic Ultrasound Fine Needle Biopsy Device (Boston Scientific, Marlborough, MA, USA) was used as the primary instrument, with needle gauges ranging from 19 to 22. In one patient, a SharkCore™ needle (Medtronic, Minneapolis, MN, USA) was utilized. Before insertion of the needle into the echoendoscope’s working channel, the inner stylet was removed, and a 10 mL syringe previously preloaded with 10 mL of negative pressure was attached to the proximal end of the FNB device (dry suction technique). The needle was then advanced under real-time EUS guidance to a few millimeters inside the target lesion. After opening the lock of the syringe to apply negative pressure, approximately 10 to-and-fro motions inside the lesion were performed, accounting altogether for one needle pass. These motions were made using a “fanning” technique inside the target lesion. Macroscopic on-site evaluation (MOSE) of the core tissue obtained from EUS-FNB was finally conducted by the operator to assess the sample adequacy.

After collection, the samples were fixed in formalin and subsequently processed in the laboratories of the Unit of Pathologic Anatomy for routine histologic, immunohistochemical, and molecular examination. Additionally, in 44 patients, a portion of the biopsied tissue sample was resuspended in 0.9% sodium chloride and processed by the Flow Cytometry Unit. The samples were labeled with a panel of fluorescent antibodies and analyzed at the flow cytometer following standard procedures.

Prior to the procedure, all the patients provided informed consent for both the biopsy and the usage of personal data for research purposes.

The exclusion criteria for EUS-FNB were the following: esophageal strictures, prior pancreatic or upper-gastrointestinal surgery, uncorrectable coagulopathy (international normalized ratio > 1.5), thrombocytopenia (platelet levels < 50,000/mm^3^), active use of P2Y12 receptor antagonists or oral anticoagulants including vitamin K antagonists and direct oral anticoagulants, active infection or fever during the previous seven days, pregnancy, breastfeeding, or the inability to provide informed consent.

The primary objective of this study was to determine the diagnostic performance of the procedure, as defined by the proportion of histologically evaluable samples and the proportion of actionable diagnoses among the total number of evaluable patients. Separate analyses were conducted for patients with and without a previous diagnosis of lymphoma. The diagnostic sensitivity to detect lymphoma was defined as the proportion of true positives (TPs) correctly identified by the test relative to the prevalence of the disease in the study cohort. Patients with insufficient material or non-pathological tissue were monitored for at least one year to assess the emergence of subsequent diagnoses. Patients without subsequent diagnosis of a neoplasia and a follow-up duration of less than one year were excluded from the analysis.

## 3. Results 

A total of 171 patients (99 male, 72 female) with a median age of 63 years (range: 19–88) underwent EUS-guided biopsies of deep lymph nodes or spleen lesions. Figure 1a,b depicts the EUS images and corresponding PET of a lymphoma patient who underwent EUS-FNB, respectively. 

The EUS-FNB procedures were not completed in 12 cases due to poor patient compliance or the absence of a target lesion that was previously identified by a different imaging technique. Additionally, 25 patients with prior non-lymphoma neoplasia were excluded. This resulted in a selection of 134 procedures for assessing the diagnostic performance (Figure 2). In 103 patients, counseling by a hematologist for suspected lymphoma or its relapse preceded the EUS-FNB procedure.

The EUS-FNB procedure was performed as both an inpatient (65 cases) and outpatient (69 cases) procedure. Spleen biopsies were exclusively performed on inpatients due to the higher risk of bleeding.

In the majority of patients (97%, 130/134), a 22-gauge needle was used with the EUS Fine Needle Biopsy Device, while a 19-gauge needle was used in 3.7% (5/134) of cases. The median number of passes during the procedures was three, with 1 or 2 passes in 15 patients (11%), 3 passes in 79 patients (59%), and 4 passes in 40 patients (30%).

The target lesion sites included mediastinal lymph nodes (*n* = 48), abdominal lymph nodes (*n* = 61), spleen (*n* = 14), and extranodal sites involving the gastrointestinal tract (*n* = 11). 

Detailed information on the specific biopsy sites is provided in Table 1.

Only two major adverse events occurred, which included gastrointestinal bleeding and a sub-capsular spleen hematoma following the EUS-FNB of spleen lesions. Both patients were managed without surgery, monitored as inpatients, received red blood cell transfusions, and were discharged after seven and eight days, respectively.

Out of the 134 biopsies, histological examination of the EUS-FNB samples produced an actionable diagnosis in 100 cases (74.6%). In 34 cases (25.4%), the histology report was considered inconclusive due to insufficient material (21 cases, 15.6%) or the presence of non-pathological tissue (13 cases, 9.7%).

The diagnostic performance did not differ significantly based on whether the biopsy site was mediastinal (79.1%; 38/48 cases) or abdominal (75.4%; 46/61 cases). The diagnostic rate for spleen biopsies was 71.4% (10/14 cases) and 63.6% (7/11 cases) for extranodal biopsies. 

The most common diagnoses were lymphoproliferative malignancies, which were found in 51 patients (37.7%), followed by solid tumors (31 patients, 22.9%) and chronic granulomatous inflammations (18 patients, 13.3%). In 13 cases, no pathological or atypical tissue was found. Table 1 provides further details on the subtypes of the diseases.

The subtypes of the diseases are reported in Table 2.

Lymphoma diagnoses were confirmed through immunohistochemical staining. Among the 51 patients with lymphoma, a specific subtype could be attributed in 47 cases (92%). For diffuse large B-cell lymphoma, the cell of origin (COO) was determined using the Hans algorithm [13,14], resulting in 12 cases identified as germinal center-derived DLBCL and nine cases classified as post-germinal center DLBCL. COO was not specified in the remaining 11 cases.

Further analysis focused on the diagnostic performance of EUS-FNB in 114 patients without a previous neoplasia diagnosis. Among them, 14 of the patients had insufficient sampled material, and 13 patients had non-pathological tissue. Consequently, the diagnostic rate among the patients without a previous diagnosis was 76.3% (87/114).

Among the 20 patients with a previous history of lymphoma, 13 relapses were diagnosed, with histological concordance in 11 cases and transformation into more aggressive lymphomas in two cases. Seven samples were not sufficient for analysis. Therefore, the diagnostic performance in patients with suspected lymphoma relapse was 65%.

In total, an actionable histological diagnosis was obtained in 100/134 of the cases (74.6%). The patients without an actionable diagnosis (insufficient material in 21 cases, non-pathological tissue in 13 cases) were candidates for a second biopsy or were observed for at least 12 months. Twelve patients were lost to follow-up, a second biopsy using a second diagnostic approach (3 US-guided biopsies, 2 lymph-node excisional biopsies, 2 splenectomies, 1 abdominal laparoscopic biopsy) was performed in eight patients, and 14 patients were observed for at least 12 months. Among the 22 patients without an initial actionable diagnosis and with a sufficient follow-up, eight lymphomas were diagnosed, either through flow cytometry (n = 3) or a second biopsy (n = 5). In the series of second biopsies, the diagnoses were diffuse large B-cell lymphoma in two, follicular lymphoma, anaplastic large cell lymphoma and Hodgkin lymphoma in one case each. 

Considering the 101 patients who underwent EUS-FNB as their first procedure and had diagnostic material obtained, a correct diagnosis was made in 98 patients (97%). The sensitivity of EUS-FNB for diagnosing lymphomas was calculated to be 86.4% (51/59).

Among the patients diagnosed with chronic inflammatory granulomatosis, four patients underwent a second sampling procedure during follow-up. In one case, a lung adenocarcinoma was diagnosed through atypical lung resection. Eight patients were treated for sarcoidosis by rheumatologists, and further work-up revealed tuberculosis in one patient. Two patients remained negative during follow-up, while five patients were lost to follow-up.

Flow cytometry analysis was applied to lymph node biopsies in 44 patients to improve diagnostic performance. In 23 cases (52.2%), flow cytometry analysis provided a diagnostic result, with agreement between cytofluorimetric analysis and histology in 18 cases (78.3%). In five cases, flow cytometry revealed an aberrant B-cell phenotype suggestive for lymphoproliferative disorder, while the corresponding histology was not due to poor sample quality. Conversely, in seven patients, histology indicated lymphoma, but flow cytometry did not due to poor sample quality or low cellularity. Figure 3 demonstrates a representative flow cytometer analysis of EUS-guided lymph node FNB.

Six patients who were initially excluded from the analysis due to poor compliance or lack of a target lesion underwent another biopsy, resulting in a definitive diagnosis in three cases: one DLBCL, one carcinomatous infiltration, and one Rosai–Dorfman disease. 

## 4. Discussion 

In this large retrospective study, we analyzed 134 consecutive patients who underwent EUS-FNB of deep lesions that were suspect for lymphoproliferative disease. Overall, the diagnostic performance was 74.6%. The diagnostic performance depends on several limitations that are inherent to the procedure. The first limitation is related to the compliance of patients during the procedure and the probability of the endosonography being able to identify the lesion of interest that has been previously identified using a different imaging technique. In our experience, this limitation resulted in the exclusion of 7% (12/171) of patients in whom FNB was not executed during EUS. Although malignant lymph nodes generally have endosonographic characteristics such as a large size, hypoechogenicity, distinct borders, round shape, and high tissue stiffness on elastography, the simple lymph node morphology, assessed through EUS, is not sufficient to distinguish benign nodes from malignant ones [15,16]. This limitation is also reflected in part by the proportion of non-pathological tissues sampled. In 13 out of 114 biopsies performed in our cohort for the suspect a lymphoproliferative disease, only normal tissue was present in the bioptic samples. 

Another limitation is represented by the quantity of bioptic material that is sampled [17]. We performed sampling with 22 G needles using a median of three passes. In 21/134 of the cases, the material was insufficient for a histopathological diagnosis. 

Recent studies have highlighted a greater diagnostic performance of the new-generation end-cutting FNB needle. In particular, a recent network meta-analysis of 16 randomized controlled trials including 1934 patients reported that the Franseen needle—provided with a crown tip with three-plane symmetric cutting edges (Acquire)—and the fork-type needle—provided with a fork-shaped distal tip including six cutting edges and an opposing bevel (SharkCore)—significantly outperformed the older reverse-bevel FNB needles for tissue acquisition of solid pancreatic masses [18]. A subsequent meta-analysis of nine studies (1276 patients) showed that the diagnostic accuracy of EUS-FNB for the tissue acquisition of suspected lymph nodes was significantly superior when performed with newer end-cutting needles (OR, 1.87; 95% CI, 1.17–3.00; *p* = 0.009) when compared to EUS-FNA [19]. These data support the promising results that emerged from our retrospective analysis, considering that all the included patients underwent EUS-FNB with newer end-cutting needles (Acquire and SharkCore).

Several studies have compared tissue sampling techniques using different sizes of needles, as well as the diagnostic outcomes of EUS-FNB and EUS-fine needle aspiration (FNA) [20,21,22,23]. One study conducted in Massachusetts involved 209 patients from five hospitals and compared the diagnostic yield of EUS-FNA and EUS-FNB using 20 G, 22 G, and 25 G needles [24]. The study reported comparable overall diagnostic accuracy between lymph node sampling using EUS-FNA combined with rapid on-site evaluation and EUS-FNB. However, the specific diagnostic accuracy for lymphoproliferative diseases was not provided. Another prospective multicenter randomized controlled trial, which included 13 EUS centers and enrolled 608 patients with solid lesions, compared the diagnostic performance of a 20-gauge FNB needle with a 25-gauge FNA needle [25]. The trial consistently demonstrated superior histology yield and diagnostic accuracy for both pancreatic and non-pancreatic lesions using the 20-gauge FNB needle. However, only a small percentage (2%) of the patients had a final diagnosis of lymphoma. Facciorusso et al. retrospectively compared EUS-FNA or EUS-FNB as diagnostic procedure for abdominal lymph node sampling in 502 patients [15]. Overall, EUS-FNB showed a higher diagnostic accuracy and sensitivity. In the group of 35 patients with lymphoma, the diagnostic sensitivity of FNB was 88.2%, which is similar to the 86.4% sensitivity in our cohort, while the diagnostic sensitivity of FNA was only 53.8% [15].

In a single-center prospective study, Hedenström et al. compared the diagnostic performance of EUS-FNA using a 25-gauge FNA needle versus EUS-FNB using a 22-gauge FNB needle in 48 patients with lymphadenopathies [11]. Among the 11 cases diagnosed as lymphoma, the sensitivity of EUS-FNB was higher than that of EUS-FNA (55% vs. 9%), although still comparatively low when considering our study and others [15,23]. These findings in favor of EUS-FNB emphasize the importance of obtaining a histological sample that preserves the lymph node architecture for accurate diagnosis and classification of lymphomas. In our case series, subclassifications based on the WHO criteria were possible in 92% of the cases.

Despite the good diagnostic performance of EUS-FNB, surgical lymph node biopsy should still be regarded as the gold standard for lymphoma diagnosis. Syrykh et al. conducted a multicenter national survey involving 31,138 cases, comparing 9924 core biopsies with 21,214 surgical excision samples [26]. Histological diagnoses were re-evaluated by expert pathologists, and the diagnostic performance of core needle biopsy (CNB) was 92.3% compared to 98.1% for surgical excision samples. Interestingly, high-grade B-cell lymphomas were more frequently diagnosed with CNB than with surgical excision, suggesting that the urgency of diagnosis may prompt clinicians to choose a less invasive and rapidly available diagnostic approach.

The survey analysis by Syrykh et al. also highlights a potential limitation of EUS-FNB in diagnosing certain lymphoma subtypes, particularly T-cell lymphomas [26]. In our case series, T-cell lymphomas were underrepresented, with only one case of anaplastic large-cell lymphoma diagnosed using EUS-FNB and a second case identified on a subsequent biopsy. Several factors may explain the absence of certain histological subtypes in our case series, including the relative rarity of T-cell malignancies compared to B-cell neoplasms, the specific localization and distribution of lymph node involvement at presentation (with selection criteria favoring deep-seated regions accessible via the gastrointestinal tract), and the histopathological challenges associated with diagnosing such diseases when working with small specimen volumes. Additional passes during EUS-FNB can provide material for flow cytometry and molecular analyses, which may be valuable in the diagnosis of lymphoproliferative diseases [27,28].

The inclusion of protein expression analysis using flow cytometry enhances the diagnostic performance and sensitivity of EUS-FNB. Particularly noteworthy is the high concordance between flow cytometry and histology, where five cases were solely identified through flow cytometry, enabling the prompt initiation of lymphoma-directed treatment in situations requiring urgent intervention. This is due to the shorter turnaround time of flow cytometry compared to immunohistochemistry. Furthermore, cytofluorimetric analysis may reduce the need for repeat biopsies [29,30]. However, the contribution of flow cytometry may be more limited in T-cell lymphomas, as these often lack specific immunophenotypic profiles [31].

Special consideration should be given to EUS-FNB of splenic lesions, as we included 14 such cases in our study. Traditionally, percutaneous image-guided core biopsy has not been recommended for studying spleen nodules due to concerns about the organ’s fragility and the associated risk of post-core biopsy complications. However, several studies have demonstrated high diagnostic accuracy of percutaneous image-guided spleen core biopsy (92–94%), particularly in cases of malignant lymphoma, with adverse event rates ranging from 2.2% to 8.2% [32,33,34,35]. A meta-analysis encompassing four studies revealed a pooled sensitivity of 87.0%, specificity of 96.4%, and a pooled major complication rate of 2.2% for image-guided percutaneous needle biopsies of the spleen, with common complications including hemorrhage and pain [36].

EUS-guided FNB of splenic nodules may serve as an alternative approach to percutaneous procedures, especially for deeper nodules, given the proximity of the spleen to the stomach walls. Lisotti et al. conducted a meta-analysis of six studies involving 62 patients to assess the safety, adequacy, and accuracy of EUS-guided tissue acquisition procedures, primarily through EUS-FNA. The meta-analysis demonstrated an overall adequacy of 93% and an overall diagnostic accuracy of 88%, with a pooled incidence of adverse events of 4.7%. Only one major bleeding event occurred in a patient with splenic pseudocysts [37]. In our cohort, EUS-guided FNB of splenic lesions exhibited a diagnostic adequacy of 85.7% (12/14) and a diagnostic performance of 91% in the cases with evaluable material. 

The EUS-FNB complication rates reported in the literature mostly concern procedures performed on the pancreas. The most frequently reported complications are post-procedural pain, episodes of acute pancreatitis, infections with fever, and bleeding [8].

In a meta-analysis of 51 different studies from a case series of 10,941 patients, the overall complication rate was low (0.98–1.72%), and the procedure-related mortality was estimated to be 0.02% [38]. Any adverse events were classified as mild, moderate or major depending on the complications [39].

No major complications were observed in EUS-FNB of lymph node sites, consistent with data from the literature [28,29,30,31]. However, it is important to note that two patients experienced hemorrhage as a complication of EUS-FNB of splenic lesions, which could be managed without surgical intervention.

The risk of the needle track seeding cells via the EUS-FNB procedure is controversial. Prospective studies that can determine the risk are currently lacking. In a large retrospective case series, the estimated frequency of pancreatic neoplastic cell spread was between 0.003% and 0.009% [8]. While this low risk is valid for solid tumors, in lymphomas, the biology of the pathology makes it systemic by itself, nullifying the risk associated with the procedure.

## 5. Conclusions

EUS-guided FNB offers several advantages over other imaging-guided techniques, such as a real-time puncture, reduced risk of complications due to the needle proximity to deep-seated lesions, and the ability to obtain samples from small lesions that may be challenging to sample using alternative methods. The high diagnostic performance, yielding actionable diagnoses in approximately 75% of the procedures, justifies the inclusion of EUS-FNB as the primary approach in the diagnostic algorithm for suspected lymphoma cases exclusively located in deep-seated sites near the gastrointestinal tract. The procedure’s short planning time, safe outpatient management, and the incorporation of flow cytometry to expedite the diagnostic turnaround time further support EUS-FNB as a rapid and minimally invasive diagnostic approach. However, it is important to acknowledge certain diagnostic limitations when dealing with difficult-to-diagnose lymphoma subtypes.

## Figures and Tables

**Figure 1 diagnostics-13-02839-f001:**
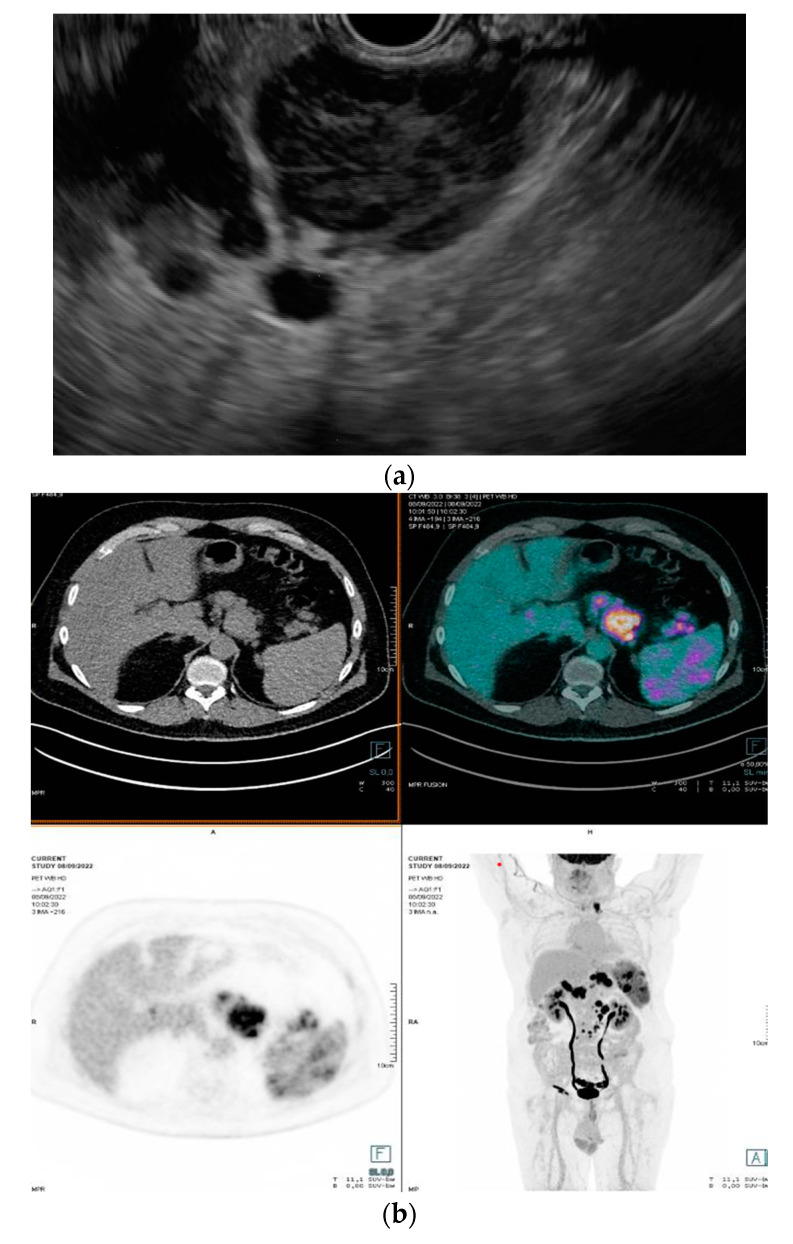
Linear endoscopic ultrasound (EUS) image of hypoechoic, confluent lymph nodes located in the periaortic region (**a**). ^18^F-fluorodeoxyglucose (FDG) positron emission tomography/computed tomography (PET/CT) scan showing an FDG-positive uptake of confluent periaortic lymph nodes (**b**).

**Figure 2 diagnostics-13-02839-f002:**
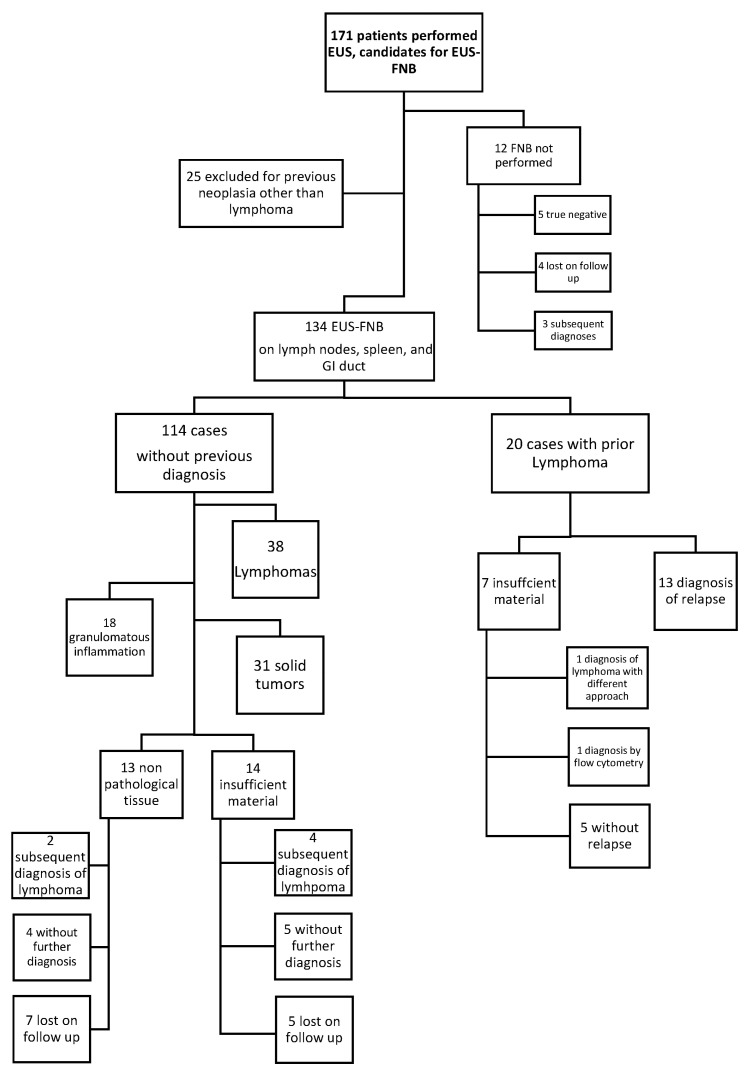
Consort diagram of the cohort.

**Figure 3 diagnostics-13-02839-f003:**
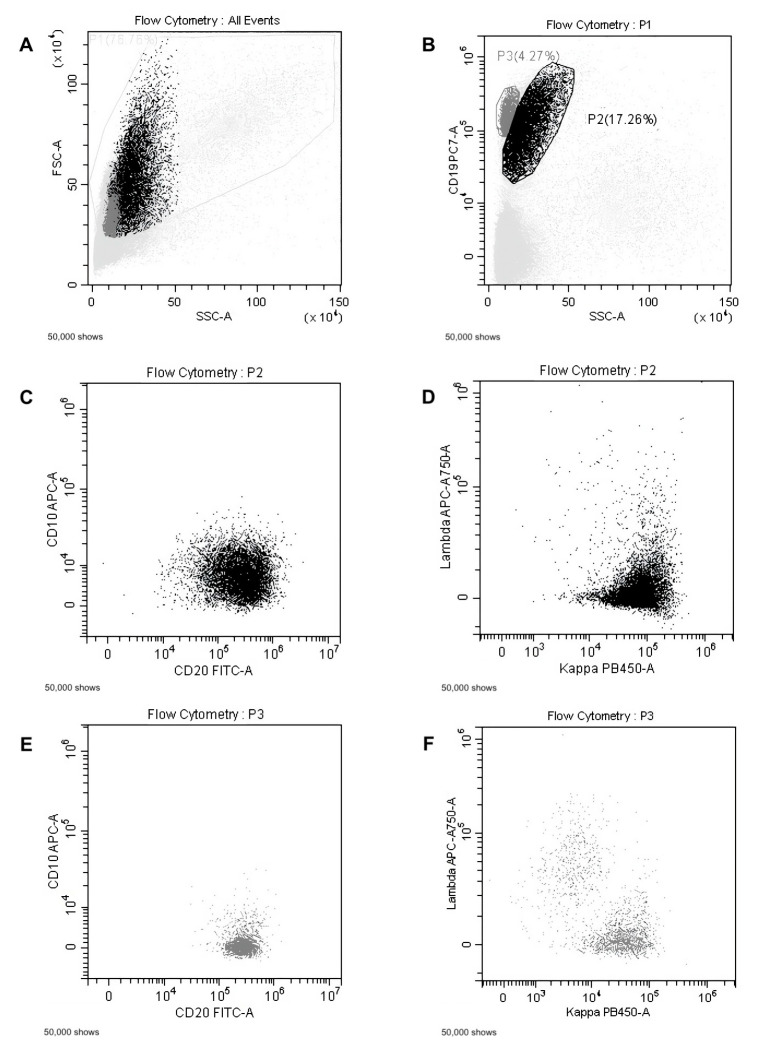
Flow cytometric analysis of an ecoendoscopic lymph node biopsy. (**A**): The total leucocyte population is gated excluding cell debris; (**B**): CD19+ population with higher side scatter (SSC) in black (P2); CD19+ population with low SSC in grey (P3). Both populations are positive for the CD20 antigen (respectively **C**,**E**), but only the pathological CD19+ population in black shows a clonal restriction for surface kappa light chain expression (**D**); while residual normal CD19+ population (grey) shows polyclonal expression of light chains (**F**).

**Table 1 diagnostics-13-02839-t001:** Sites and number of biopsies in 134 EUS-FNB.

Site of Biopsy	Number of Biopsies	Diagnostic Rate
Mediastinal lymph node	48	79.1% (38/48)
Sub-carenal	21	
Posterior-mediastinal	16	
Other mediastinal nodes	11	
Abdominal lymph node	61	75.4% (46/61)
Perigastric	6	
Periduodenal/jejunum	5	
Hepatic hilum	6	
Splenic hilum	3	
Peripancreatic	4	
Paraortic	1	
Pelvic	1	
Not further specified abdominal nodes	35	
Spleen	14	71.4% (10/14)
Extranodal lesions	11	63.6% (7/11)
Esophagus	1	
Stomach	4	
Duodenal/jejunum	2	
Pancreatic	2	
Liver	1	
Peritoneal nodal	1	

**Table 2 diagnostics-13-02839-t002:** Diagnoses using EUS-FNB in 134 cases.

Diagnosis	Number of Patients
Lymphoproliferative diseases	51
Diffuse large B cell lymphoma	32
High grade B cell lymphoma	2
Follicular lymphoma	5
Hodgkin lymphoma	5
Indolent non-Hodgkin lymphoma	3
Anaplastic large cell lymphoma	1
Mantle cell lymphoma	1
Marginal zone lymphoma	1
Non-Hodgkin lymphoma NOS	1
Solid malignancies	31
Ovaric adenocarcinoma	1
Lung adenocarcinoma	7
Lung squamous carcinoma	2
Small-cell lung cancer	1
Pancreatic adenocarcinoma	3
Squamous carcinoma	2
Inflammatory myofibroblastic tumour	1
Mesenchymal neoplasia	1
Hepatocellular carcinoma	2
Cholangiocarcinoma	1
Gallbladder cancer	2
NET	4
Gastric adenocarcinoma	2
Prostate adenocarcinoma	1
Papillary thyroid carcinoma	1
Chronic granulomatous inflammation	18
Not pathological tissue/not atypical	13
Non evaluable	21

## Data Availability

The data presented in this study are available on request from the corresponding author. The data are not publicly available due to confidentiality and ethical reasons.

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
