# Peer review of "Endoscopic Ultrasound-Guided Fine Needle Biopsy in the Diagnostic Work-Up of Deep-Seated Lymphadenopathies and Spleen Lesions: A Monocentric Experience"

_diagnostics, 2023, doi:10.3390/diagnostics13172839_

Round 1

Reviewer 1 Report

This study evaluated the diagnostic utility of fine needle biopsy of deep lymphadenopaties and spleen lesions. It is a single centre study and the authors report a sensitivity of FNB for diagnosing lymphomas to be more than 80 %. I have the following comments:

1.       In the introduction section the authors should discuss possible complications of FNB

2.       Does FNB increase the risk of malignant spread in certain tumour types?

3.       Why did the authors choose a period from June 2017 to December 2021?

4.       What were the exclusion criteria?

5.       The procedure of FNB should be described in greater detail

Reviewer 2 Report

Very interesting and well conducted study. My comments:

1) Sample size is good so no specific comment on that

2) The authors should comment on the important role of newer end-cutting FNB needles in EUS-guided tissue sampling (cite PMID: 35124072)

3) Follow-up of less than 1 year is not enough to represent the gold standard for accuracy in non-pancreatic lesions (check the AGA white paper)

Round 2

Reviewer 2 Report

The revised version of the manuscript is OK. Thank you!